# “If It Goes Horribly Wrong the Whole World Descends on You”: The Influence of Fear, Vulnerability, and Powerlessness on Police Officers’ Response to Victims of Head Injury in Domestic Violence

**DOI:** 10.3390/ijerph18137070

**Published:** 2021-07-02

**Authors:** Jenny Richards, Janet Smithson, Nicholas J. Moberly, Alicia Smith

**Affiliations:** 1Somerset NHS Foundation Trust, Taunton TA2 7PQ, UK; 2Department of Psychology, University of Exeter, Exeter EX4 4QG, UK; J.Smithson@exeter.ac.uk (J.S.); n.j.moberly@exeter.ac.uk (N.J.M.); A.Smith@exeter.ac.uk (A.S.)

**Keywords:** domestic violence (DV), health consequences, psychological consequences, head injury, traumatic brain injury (TBI), police

## Abstract

Domestic violence (DV) victims face significant barriers to accessing healthcare. This is particularly concerning in cases of brain injury (BI), which is difficult to diagnose and risks severe long-term consequences for DV victims. Police may be able to identify head injury (HI) and signpost victims to healthcare. This research investigated potential barriers to police supporting victim health needs by exploring police attitudes towards DV and considering how police interpret and respond to stories of HI in DV victims. Individual interviews were conducted with 12 police officers from forces in South and Central England. This included the use of a clinical vignette. Thematic analysis highlighted three global themes: ‘seesaw of emotions’, ‘police vulnerability’, and ‘head injury is fearful’. Police officers’ vulnerability to external blame was the predominant influence in their responses to HI.

## 1. Introduction

Domestic Violence (DV) is “any incident or pattern of incidents of controlling, coercive, threatening behaviour, violence or abuse between those aged 16 or over who are, or have been, intimate partners or family members regardless of gender or sexual orientation” [1] (p. 5). The Office for National Statistics [2] suggests prevalence rates for DV in England and Wales are between 4.2% and 7.9%. DV victims are often isolated from networks of support, and face barriers accessing healthcare [3,4]. This is a particular concern in cases of mild traumatic brain injury (mTBI), thought to be common in DV, which can have serious long-term consequences, and is routinely under/misdiagnosed [5]. Police officers regularly interact with DV victims immediately following violence and could identify and support victims with mTBI [6]. This study investigated officers’ attitudes and responses to stories of head injury (HI) in DV in order to explore potential avenues of police support for DV victims with mTBI.

### 1.1. Domestic Violence and Health

DV can have wide-ranging effects resulting in specific problems for individuals’ physical health, sexual health, and mental health. Consequences range from minor injury to chronic conditions or fatality [7,8,9]. Repeated physical assaults to specific areas of the body can lead to long-term damage, such as with injuries from repeated non-fatal strangulation (NFS) and HI [10]. The head, neck, and face are the most commonly injured body parts in DV [11]. Brain injury, particularly mTBI, is thought to a common consequence of DV [12]. Diagnosing mTBI in DV victims can be challenging as this population may be unlikely to seek help for health issues, with barriers including difficulty disclosing DV, shame and low self-esteem [4,13,14,15].

MTBI can have a lasting impact on everyday functioning including impairments in memory, concentration, attention, and problem solving [16]. These injuries could have a dangerous effect on DV victims. Impairments in memory, judgement, and decision-making could make victims more likely to return to perpetrators, leaving them vulnerable to further abuse [17,18]. When police are called to DV incidents, they have a unique opportunity to identify health issues and promote healthcare access. Police officers’ ability to take this role is important as victims are unlikely to seek support independently, and rarely judge that their injuries justify medical intervention [19].

### 1.2. Police Attitudes toward DV

Despite increasing recognition of DV as a criminal offence, there are ongoing barriers to police supporting victims [20]. One key barrier is police attitudes to DV. Police can have difficulty understanding the complexity of DV and struggle to identify abusive behaviour, which can result in officers blaming victims for remaining with abusive partners [21,22]. Newly developed DV training programmes have been successful in improving officers’ understanding of DV, but have had limited success addressing underlying attitudes [23,24]. Addressing officers’ attitudes is important to the success of police reforms [25]. Furthermore, attitudes toward specific groups shape views of policies that affect the welfare of those groups [26]. Attitudes towards DV could influence how police respond to victims, and their beliefs about expanding their role to support victim health.

### 1.3. Attitude Theory

Conceptualisations of attitude generally support an umbrella definition: “a psychological tendency that is expressed by evaluating a particular entity with some degree of favor or disfavor” [27] (p. 1). Beyond this, researchers disagree on where attitudes sit on a spectrum between stable, consistent entities [28,29], and temporary constructions [30,31]. The contrasting ends of this spectrum align with a wider difference between positivist and constructionist ontologies. Historically, attitude theory has complemented a positivist approach: a decontextualised perspective in which an individual responds to attitudinal objects [32]. Attitudes have traditionally been measured by positivist quantitative approaches which focus on quantifiable empirical evidence [33]. These include explicit measures (e.g., self-report scales) which assume introspective access to attitudes [34], and implicit measures (e.g., implicit association tests) which capture attitudes without requiring participants to report a subjective assessment [35]. These positivist quantitative approaches can provide a snapshot of attitudes, relevant to the specific context and moment of measurement [36]. This approach tends to align with the concept that attitudes are quantifiable, stable entities [37]. 

Social constructionist scholars have challenged the traditional conceptualisation that attitudes are enduring, quantifiable entities [38]. Social constructionists emphasise the variability in attitudinal responses across/within cultural, historic or social contexts [30,39]. From a social constructionist perspective, attitudes are best understood as being constructed from a socialised notion of the self [40]. The emphasis is that meanings associated with attitude objects are largely socially constructed, and therefore attitudes need to be understood within their social context [41,42]. Central to social constructionist theories is the belief that human knowledge of the world is constructed through language, culture and context [43]. Attitudes are negotiated and situated in particular historical circumstances, and form part of interactive and dynamic relationships between social knowledge, social identities and relationships [44]. In comparison to quantitative approaches, qualitative investigation can explore aspects of attitude that cannot be quantified, often focusing on the influence and interactions of social contexts.

Policing is a social activity entrenched in symbolic tools (e.g., uniform, rank) that people use to describe and discuss their social environment, and their identity within it [45]. Furthermore, many studies of the police reference police culture: a collective sense-making where beliefs, attitudes, and behaviours are influenced by social relationships with and within the police organisation [46]. During the process of this research, protests against police brutality, racism, and violence erupted across the world [47]. Protesters drew attention to problematic police culture, discrimination, failures to protect vulnerable people, and the persecution of specific groups.

### 1.4. Impact of Austerity on Policing DV

It is relevant to consider the context of austerity in the UK which may further influence group identification, outgroup derogation and police attitudes to DV. The global financial crash of 2007–2008 led to over a decade of austerity policies, putting vast economic pressures on public services including police [48]. Between 2010 and 2019, real-terms government funding for police fell by 19%, with an 18% reduction in police workforce [49]. Cuts to funding have resulted in fewer officers, closures of police stations, and organisational restructuring [50]. Budget cuts have had adverse consequences for crime reduction [51] and the protection of vulnerable people, including DV victims [49]. 

The pressures of austerity policing appear to have increased discontent, led to a reduction in progressive policing (e.g., community/relationship oriented), and a fall back to traditional attitudes including distrust toward the public, a crime-fighting focus, and loyalty to fellow officers [52]. Increased criticism and depleted resources may have increased hostility towards outgroups (e.g., the public, offenders, victims).

Police could have an important role in supporting DV victims by identifying health needs, signposting, and empowering victims to access health services. This is of particular importance in mTBI, where symptoms are often missed. Better understanding of police attitudes towards DV may support the development of more effective training and improve police responses.

### 1.5. Present Study

To our knowledge, there has been no research exploring police responses to the health needs of DV victims. The proposed research aimed to investigate police officers’ attitudes and responses towards victims of HI in the context of DV. Critically, this could help better understand barriers to police supporting DV victims in accessing healthcare and psychological support.

The objective of this study was to capture the opinion of the perspective of police officers. The study was conducted as part of a doctoral qualification by the first author, and was guided by the following questions:How do police officers construct attitudes towards victims of DV?How do police interpret and respond to stories of head trauma or symptoms of BI with victims of DV?

## 2. Materials and Methods

### 2.1. Research Design

Semi-structured individual interviews were selected as the method of investigation. Although focus groups were another possible method, they are subject to group dynamics [53]. It seemed likely that this sensitive topic would be better studied through individual interviews [54]. Semi-structured interviews offer flexibility around predetermined topics, encouraging investigation into specific research questions, whilst allowing for the discovery of unanticipated topics [55].

### 2.2. Sample

Five police forces were contacted with details about the study. Two police forces in Southern and Central England participated. Participants were recruited with the assistance of police liaisons who circulated details about the study via email. Officers who wished to take part contacted the researcher directly. Some officers were selected to participate by a senior officer based on availability.

In total, 12 police officers volunteered between October 2019 and March 2020. Participants were required to be aged at least 18 years and a serving police officer of any rank.

### 2.3. Procedure

Interviews took place in a private room within local police stations and were recorded using a dictaphone. An interview schedule was developed by considering the research questions within existing literature. Questions explored participants’ experience of responding to DV, attitudes, knowledge and understanding of DV, knowledge of HI and mTBI, experience of training in DV and HI, and attitudes towards the police role in DV/public health. A clinical vignette was used to promote discussion about responses to head trauma. The vignette was based on Markowitz and Watson [56], and briefly described an incident of DV involving a woman who had been assaulted to her head with varying features of clinical interest (visible HI and symptoms of concussion).

A pilot interview was conducted with a police officer to assess the suitability, language, order of questions, and the readability of ethics documents [57]. This pilot also enabled opportunity to practice interview skills and reflect on interviewer style. Interviews ranged between 36 and 95 min. Interviews were transcribed prior to analysis.

### 2.4. Analysis

Data were analysed using thematic analysis (TA): a method of exploring, identifying, analysing, and reporting patterns or themes within data [58]. TA is an active process where the researcher becomes the instrument for analysis: making judgments about coding, theming, decontextualising, and recontextualising the data [59]. The creativity and position of the researcher is an integral part of analysis [60].

Data were analysed based on the six-stage framework proposed by Braun and Clarke [58]. Data were uploaded to NVivo to support the process of analysis. The first phase involved immersion in the data by listening to audio recordings, reading and rereading transcripts, and making notes of early theme ideas. The second stage involved line-by-line coding. This study took a deductive approach to TA: codes were initially organised according to the interview schedule and considered within existing literature. New and novel codes continued to emerge from the data during line-by-line coding. The third stage involved organising codes into over-arching “global themes” and ‘nodes’ on NVivo. The fourth stage involved reviewing and re-applying themes to ensure they represented the data. This was an iterative process with themes modified, added, and deleted throughout.

Phase five involved defining and naming themes and reporting the data in writing. Phase four and five of TA were integrated with the rectification stage of TA [61]. Rectification consisted of three stages. First, a distance from the data was maintained, before returning with a fresh angle to support a self-critical approach. This stage was additionally supported by discussing the themes with researchers within and external to the project. The second stage involved relating themes to established knowledge, consulting theory to develop links. Finally, in the stabilizing stage, themes were described with divergent views to challenge generalisations. 

Strategies employed within the analysis process included creating visual displays (tables, flow charts, thematic diagrams), and maintaining a reflective journal. 

### 2.5. Credibility Checks

In order to establish trustworthiness, a number of credibility checks were used to ensure the analysis provided a fair interpretation of participants’ views [62]. There was prolonged engagement with the data including reading and rereading transcripts alongside audio recordings. Data triangulation was utilised by recruiting from police forces in separate areas (rural and urban) and including officers of differing ranks and departments. Peer debriefing provided an external check on the research: transcripts were shared and discussed with a qualitative research group, who helped promote reflective thinking and limit bias. Finally, regular contact with supervisors to discuss emerging themes in relation to researcher emotions/interpretations helped separate the data from the researcher to ensure accuracy in analysis.

### 2.6. Researcher Reflexivity

I (the first author and interviewer) am a liberal feminist, with personal and professional experiences of DV, and limited experience with police. My gender, occupation, and age group were identifiable to participants visually, and through our correspondence, and may have influenced our interactions. Strategies used to consider my assumptions and biases as a clinician and researcher for this topic included reflecting on my multiple identities, reviewing transcripts for inappropriate clinical style, and piloting interviews. Debriefing and attending to my influence while analysing the transcript were important to promoting impartiality. Debriefs were supported by psychologically aware peers and my feminist supervisors.

## 3. Results

To address the research questions, three global themes were identified in the data, with various sub-themes; these are expanded on below, and illustrated with relevant quotations from interviewees. In response to the first research question, “How do police officers construct attitudes towards victims of DV”, the global theme of ‘seesaw of emotions’ was developed. Analysis suggested that participants held conflicting attitudes towards victims, shaped by their experiences of powerless, anger and frustration when responding to DV. This created uncertainty around victimhood, where police attitudes were a mix of sympathetic, supportive, and blaming.

In response to the second research question, “How do police interpret and respond to stories of head trauma or symptoms of BI with victims of DV”, two global themes were identified. The theme ‘head injury is fearful’ demonstrated that officers found the unpredictability of HI anxiety-provoking, often responding with caution. The theme ‘police vulnerability’ reflected police feeling scrutinized, underequipped and under-resourced. Officers’ responses to HI in DV seem driven by their own vulnerability to blame.

Figure 1 is a thematic map created by the authors to demonstrate how sub-themes were organised under global themes. This figure also highlights the interconnectedness between sub-themes in the analysis. For example, ‘scrutiny and blame’ fell under the global theme ’police vulnerability’ but the data were also connected with the sub-themes ‘powerlessness’ and ‘better safe than sorry’. A thematic table with supporting quotations was developed.

### 3.1. Global Theme 1: Seesaw of Emotions

This global theme captures how officers’ experience of powerless and frustration influenced the construction of their attitudes towards DV victims. This emerged from three sub-themes, which are described below.

**Subtheme 1:****Powerlessness.** Complex relationships between police and victims were described. Officers reported a desire to help yet relied on victim support which was often refused.


*I’d probably say at least 75% we go, arrest the offender, male or female, and then they don’t want to press any charges or make a formal complaint, give us a statement, support the prosecution.*
*(Rachel)*


*There’s only so much I can do as a police officer, which is hard because I want to do more sometimes. But I can’t make you leave that abusive partner. As much as you will sit there... I’ve sat there for hours with victims trying to get them to make that first step. Sometimes they’re the one who’s got to make that leap.*
*(Ian)*

All participants discussed the limitation of their role in DV.


*We are not the organisation that people want to see because we are of no help to them whatsoever. We can’t direct refer, we can’t offer any clinical advice, we can’t offer any help whatsoever.*
*(Liam)*

All participants mentioned wanting to support victims but suggested they were ‘not the service’ for DV, referring to their lack of training, ‘intimidating’ uniform, and perceived inability to provide long-term solutions.


*You don’t have any training as a police officer to actually speak to someone that might be suffering a crisis. (Paul) Unfortunately, with the police officers, it’s a lot of putting a plaster on at times. We can deal with fixing the short-term.*
*(Thomas)*

**Sub-theme 2: Anger and frustration.** All participants described frustration linked to repeat victims/offenders, low prosecution rates, and excessive paperwork in DV. Officers indicated a mutual distrust of victims and some expressed frustration and confusion toward victims. 


*It gets frustrating, because you feel like, “We’ve done our best to help you, and you’re still at risk, and it looks as though you’re putting yourself at risk.”*
*(Jack)*

Officers struggled to understand DV, particularly why victims stayed, and why perpetrators were violent. 


*I don’t get it, if I’m honest. I get a lot of things in policing; I understand why people steal things, I understand why people potentially become sex offenders—I can see the mind-set, and why people get caught up in that. I struggle with domestic violence, because on a personal level, I just don’t get it really.*
*(Ian)*

Some participants offered concrete reasoning of why victims might stay in relationships but struggled to relate.


*They stay for years, some of them. I understand it is really difficult for them to break free, but I can’t process why it is that difficult for them to break free.*
*(Will)*


*I think it is sad in a certain extent. I think there are occasions when things are really bad where you think, I can’t think logically as to why that person would do that. Sometimes you get frustrated with that person, particularly when you are trying to help them and they don’t want to help themselves.*
*(Liam)*

Some officers were not just frustrated, but angry. Anger was usually directed to perpetrators of DV who were described as “heinous”, “despicable”, and “narcissistic, self-absorbed sociopaths”. Difficulty managing anger was apparent in responses. For example, when asked how he felt towards perpetrators of DV Ben replied:


*Our job is to be impartial, it’s not to be judge jury and executioner… My feelings towards D… DV perpetrators myself. I hate them. I think they’re horrible people, men and women alike… My thoughts to people I nick for domestic violence. I don’t get emotionally involved because that’s not my job.*


Sometimes anger was expressed in the interview room with raised voices or facial expressions. One participant described violent fantasies towards perpetrators of violence. 


*It’s frustrating when you walk into addresses and you’ll see the same people who have injuries, and you look at the bloke or the female and they’re almost cocky with it. ‘What are you going to do?’ And you just think, if I could just have five minutes, I’d kick seven bells of shit out of you, and see how you like it.*
*(Ian)*

The perceived mundanity of DV contrasted with anger. According to participants, DV is their “bread and butter”, “our volume business”.


*With quite a lot of domestics, you feel like a taxi service.*
*(Paul)*


*A vast majority of our calls are domestic related. And obviously, the ones that stick out in your brain are the ones that usually are significant because something horrific has happened. But actually, I’ve been to equally many that weren’t horrific, but you don’t remember all the details and all the things of those because you are going to several a day.*
*(Emily)*

Incidents were described as “run-of-the-mill” and “routine”. Officers discussed the processes and paperwork.


*It can just be really mundane, and it’s like, “Yes, you’ve had an argument, okay, here’s the process,” and you’re not thinking, you’re not challenged, and you’re just like, oh, it’s just process. No, I don’t like them.*
*(Jack)*


*You know that there is an amount of paperwork that goes with that so there is always an amount of mundanity to it if it is relatively routine.*
*(Liam)*

Frustration and powerlessness appeared to be mitigated by rare occasions of positive feedback in DV. 


*Just by chance, two years ago I was doing just a traffic job up on X Street. She came out of the shop and said, “I remember you,” told me all about it, and just said, “I’m just so glad that I had the courage to come down and talk to you that night, and the police supported me through everything.” And that was like, well, if that’s just one person, great.*
*(David)*

**Sub-theme 3: Victimhood is uncertain****.** Participants’ responses suggested they held concepts of good or bad victims and encountered situations where they struggled to identify a clear victim. 


*There is no not genuine victim ever—but it’s the ones where it’s messier and harder to work out who’s more to blame, or who’s not to blame, or what’s gone on*
*(Emily)*

Participants spoke with compassion and empathy; although, at times this was layered with frustration.


*There is a blame culture but there’s also empathy you know, and we do feel genuinely sorry for people. We do feel genuinely worried for people. We also feel very worried for people who actually aren’t willing to help themselves. There’s some people who you just know you can help that night and you know tomorrow there gunna have them back and they’re going to have the living crap beaten out of them next week.*
*(Ben)*

Eleven of the twelve participants spoke about “types of victim”. Responses fit within the existing literature [63,64] and described four types of victim.


**The “genuine victim:”**



*You do get different jobs where you can see which one is a genuine victim and you feel sorry for, and you’ve got to try and not feel sorry for and just help them.*
*(Jack)*


**The “manipulative victim”:**



*Sometimes I think there are occasions where you have victims that may well be offenders on different days or may well play up to stuff and use what they know will happen to domestic abuse suspects to make up allegations.*
*(Liam)*


*A lot of the time it’s because between couples, they use us, they know that they can ring the police and if we turn up and they say, “I want them out of my house,” then it’s as if we are there like security guards kind of thing and we do get used for that quite a lot*
*(Rachel)*

**The “one-off victim”** where the perpetrator held less blame: 


*Sometimes, arresting people just because it is a domestic doesn’t necessarily mean it’s the right thing to do. It might be the first time that anything’s ever happened between them, and someone’s lost their temper because of some trivial matter and someone’s ended up getting hurt, which isn’t right, but it doesn’t mean that by taking away their liberty and arresting them.*
*(Paul)*

**The “just as bad” victim** where there was mutual violence:


*You get sometimes people who are highly dysfunctional. One of the big taboos about domestic violence is mutually abusive relationships, and they do happen, where you get… it sounds almost dismissive to say they’re both as bad as each other, and it’s not meant to, but unfortunately there are situations which are like that.*
*(Michael)*

Victim typology was linked to police involvement. Those who accepted police help were deemed ‘genuine’.


*It’s going to sounds terrible to say but um its true, um, but there is a real victim and a victim for the sake of being a victim. So, we have our real victims, the ones that suffer in silence. The ones that suffer abuse day in day out for a period of time. That hide it from their family, that hide injuries, that hide psychological injuries, who eventually have the bravery to come forward or approach someone and want our help.*
*(Ben)*

All but one participant discussed specific stereotypes about victims, referencing drug use, alcohol abuse, and mental health. 


*Most of them are. I’d say most are around drugs and alcohol.*
*(Rachel)*


*Um… but a lot are (sighs) you know I think a lot what we classify as domestic abuse is very closely intertwined with mental health, substance abuse and history of abuse themselves.*
*(Ben)*

When discussing victims, participants often took care to use gender neutral terms (e.g., they, victim, offender). When pronouns were used these were always female in reference to victims, with male pronouns to describe offenders.


*Stereotyping a bit, but generally it is male perpetrators against women.*
*(Will)*


*Most of the stuff, to be fair, that I go to is arguments and it’s women who may have an injury to an eye, a black eye.*
*(Peter)*

One example was given of DV in a same sex relationship, and one with a male victim/female offender. Issues relating to inequalities in gender and sexuality were only discussed by one officer who worked in the specialist DV team.


*Anyone who’s been a police officer longer than five minutes knows it’s a mistake to automatically assume that the man’s the aggressor and the woman’s the victim.*
*(Michael)*

Participants expressed a desire to protect and safeguard victims. All officers mentioned the complexities of DV, but still struggled to relate to the victim’s experience.


*It might be that they can’t actually leave each other. It might be that they’ve tried to leave and it made it even more violent, or they have left and they’ve found them again, or they’ve left and they just can’t cope being on their own. Some people have got poor mental health, and it might not be like mental health in terms of an actual illness, but it might be low confidence, anxiety; they can’t be on their own. Sometimes you go to the same people, but it’s a different partner, but they’re still being offended against. And it’s difficult to understand why; if you’ve removed the offender and you’re no longer with him or her, and you’re in a new relationship, why is this still continuing?*
*(Jack)*

Participants struggled with victims who refused to cooperate with prosecution, but still called for help. Narratives around victimhood were interwoven with feelings of powerlessness.


*A lot of people see that as a trouble when they go to the incidents in the first place, especially if it’s a recurrent address they’re going to and you know what the result will be before you get there, where they’re likely to not talk to you, not provide a statement, and if you arrest this person and take them away, they’ll just be coming back a few hours later and probably doing the same thing again. And a lot of people, they can’t help themselves; the police will just try and help them and help them, but if they don’t want the help, it’s not going to help them.*
*(Paul)*

Emily spoke about the difficulties challenging her peers on negative language or attitudes.


*The problem is, you don’t always have time to do it, and there’s a lot of supervisors who won’t prioritise having that conversation. I did, because it was something I was passionate about, and I hated hearing that black humour; I hated hearing that desensitisation and those coping strategies. I understood them all, but I just didn’t like them. So, I would have those conversations. But actually, like I say, I had to be very sensitive and careful what times, because you could feel your staff withdrawing, “Oh god” and rolling their eyes, “not that conversation again”.*
**

### 3.2. Global Theme 2: Police Vulnerability

**This second global theme** highlighted participants’ struggle to meet the demands of their role and fear of blame. Three main sub-themes for this are outlined below.

**Sub-theme 1: Under-equipped and under-resourced.** All participants portrayed police forces as stretched. Officers described how their role had changed to include social care, mental health, public health, and DV. 


*We are dealing with so much more—for instance, mental health—we are dealing with so much more of that now than we did even ten years ago, because those services have been stripped back. And we don’t want to end up in a situation where basically, and this is already going a lot, where police are picking up the shortfall of other departments.*
*(Michael)*

Ten officers felt comfortable attending a range of incidents, highlighting part of their role is ‘preservation of life’. However, they felt they did not have the training or resources to be effective. 


*Sometimes we are paramedics, stabbings, shootings, whatever else that we go to, mental health, like I say, paramedics, psychiatrists, going out to children, child services, as well as trying to catch criminals. There’s only so much that we can do.*
*(Rachel)*

Although these participants accepted a dynamic role, a minority were angry about the changes in their role.


*The problem is, as I say, because we deal with the criminal side, if you take on too much responsibility over the emergency services side, we are going to be more paramedics from then on, rather than the police officer.*
*(Thomas)*

Participants who felt their role should be limited to managing crime were also those who expressed the most anger toward policing DV. 


*I would be pushing the other way to say that is not the role for the police at all. The police have a defined role and if you are going to cloud that role then you detract from other areas of society. The societal expectation is that police deal with crime and bad stuff.*
*(Liam)*


*I think that our job should be to deal with criminality. I’m sick to death of mediating crap.*
*(Ben)*

It was apparent in every interview that officers were struggling with increasing demand and decreasing resources. 

**Sub-theme 2: Blame and scrutiny.** Apparent in all interviews were participants’ experiences of scrutiny, particularly in DV, which impacted their actions and relationships with victims. 


*People will go against the victim’s wishes, because if they don’t do that, the 9 o’clock jury will have a go at them, and if something happens, they’ll be responsible for it. So, you’re damned if you do, damned if you don’t. It’s almost like that blame culture.*
*(Ian)*


*When I’m in work there’s very much a blame culture so it’s constantly that, there’s that train of thought that (pause) if this isn’t done properly, I’m right up the creek without a paddle. So, there’s that constant fear that if you don’t get it right then it gunna be you in front of the coroner or you in front of a disciplinary board.*
*(Ben)*

Many participants displayed anxiety around the interview making comments such as ‘*it sounds bad but…*’ or ‘*I shouldn’t say this but…*’. In the below exert Will hesitated in responding, stating ‘I’m conscious I’m stereotyping people in a recording’. The interviewer sought to reassure him:*I*:*It is all anonymous.**R*:*‘Police spokesman was heard to say...’ That is what the press do!**I*:*I am not looking to catch you out, I am just interested in your…**R*:*That is fine. You just get in this defensive mood whenever you are being asked an opinion. I have got to be really careful what I say because the media tend to take out all the context and just keep the punchline.*

**Sub-theme 3: You cannot train for emotions.** Participants described feeing unprepared for the level of violence and injuries they encountered, stepping into the emotional world of the public, and their own emotional reaction.


*Until you physically come out and you do the job, you have experienced it, you take on the emotion of the people involved, you’re getting the emotional side, people crying, screaming, erratic ones who get to each other. That is something that you can’t train for in that sort of classroom environment.*
*(Thomas)*

Nine officers described awkwardness around policing DV, particularly in the intimacy of relationships. 


*I don’t like it. I don’t like it at all. I’d much rather go to a theft, or anything else, really. Domestics, they’re so tricky; they’re so difficult and they’re so familiar.*
*(Jack)*

While participants could generally separate themselves from victims of crime such as theft, the familiarity of relationships in DV left them struggling to detach. This may remove the layer of protection officers experience in identifying as separate to ‘civilians’. Throughout interviews, participants demonstrated a pendulum of relating to and distancing from DV victims. 


*It is very easy to say, ‘Why are you with that person? Why would you go back there?’ It is very easy to say that and from a logical perspective you would always say, ‘Why would you do that?’ but you can see why people do that because everyone has been in relationships where things aren’t great—not necessarily to that standard. Things aren’t great, but you persevere.*
*(Liam)*

Eleven participants discussed becoming ‘desensitised’ and cutting off emotionally, which was described as ‘necessary’. 


*I think it…(sigh)…it depends, from a…from a policing perspective I’ve done this job for a long time and I’ve become almost immune to is and I disassociate myself from my profession out of work.*
*(Ben)*

However, there were descriptions of emotional experiences; three officers recounted experiences of abusive relationships themselves, or within their family, and two described trauma symptoms related to policing. 


*I mean, if you want horrific stories, then yes, I’ve got a catalogue of them in my brain that I sleep with every night*
*(Emily)*

Peter described being assaulted while attending a DV incident:


*You put it in your box, there’s always a box for it, and you put it away. And I couldn’t with that job, so I had a little bit of counselling. And that helped, I got through it, so I’m fine now from that.*
*(Peter)*

### 3.3. Global Theme 3: Head Injury Is Fearful

The final global theme addresses police interpretation of HI in DV. Participants were confident managing first aid but anxious about the unpredictability of HI. Two main sub-themes were developed within this global theme:

**Sub-theme 1: Head injury is unpredictable****.** Officers had differing knowledge and interest in HI. No participant had heard the phrase mTBI, but many were aware of the signs and risks of HI. Two officers had incorrect information and did not believe strangulation could lead to BI. Five officers believed injuries to the face were not HI.


*A bang on the back of the head is going to be more dangerous than a hit to the face, because that’s where your brain is, isn’t it?*
*(Jack)*

Eleven officers focused on visible injuries. When discussing mTBI, six participants showed disinterest, suggesting this information was not relevant to their role. Three talked about the impact on evidence gathering, and two were interested in how mTBI may impact interactions with victims. No officers made links between mTBI and victim’s vulnerability to DV.

Another commonality was fear of HI. Eleven participant recounted stories, experiences or examples in the media of an untreated HI that resulted in a fatality.


*A bang to the head’s very serious; it will kill you. It might not even kill you there and then, it might kill you afterwards.*
*(Jack)*


*We have got a pathological fear of head injuries as an organisation*
*(Will)*

This seemed particularly worrying to officers in DV as they relied on victims to give an accurate report of injuries. 


*The one thing that can make it a bit difficult is that the person who’s received the injury can sometimes be reluctant to talk about the mechanism of injury.*
*(Michael)*

Participants also spoke about the reluctance of victims to attend hospital. Ten participants called paramedics regardless, whilst two accepted the victim’s choice. 


*It all comes down to what policing power we have and we don’t have the power to drag people to hospital and say, “You are going”.*
*(Rachel)*

**Sub-theme 2: Better safe than sorry.** Police officers reported that their anxiety about HI led them to ‘err on the side of caution’. All participants said that they consulted paramedics if they suspected a HI. 


*I think you’ll find that with us in general, because of the nature of the head as such, and if anybody mentions headaches, ringing in their ears, feeling a bit fuzzy or whatever, we’d call somebody.*
*(David)*

When discussing decisions to call paramedics, participants focused on having a medical professional take responsibility for decision making. Very few participants mentioned ongoing concern for the victim’s health. This is not to suggest that officers are disinterested in the safeguarding of victims, as many spoke about this as a key part of their work. In participants’ accounts, fear of scrutiny and blame appeared at the forefront of their decision making. 


*You wouldn’t dream of leaving people with a head injury, just in case. We operate very much on that ‘just in case’, because if they did and they died, you are looking at job-losing territory.*
(Ian)


*If it goes horribly wrong the whole world descends on you.*
(Will)

## 4. Discussion

This study aimed to explore police attitudes towards DV victims, and to understand how police interpret and respond to stories of HI in DV, in order to provide more effective training, support, and interventions for both police and victims of DV. Analysis suggests police hold conflicting attitudes toward DV victims, shaped by experiences of powerless when faced with injured victims in tense and intimate disputes. While police expressed sympathy and a desire to help DV victims, they also conveyed frustration which was exacerbated when they felt unable to fulfil their role because victims did not disclose injuries, refused perpetrator arrest, or declined medical help. Ten participants expressed attitudes that suggested they placed some responsibility or blame with victims for ongoing violence (e.g., “they don’t help themselves”, “they’re not changing their lifestyle”). These attitudes towards victims appeared to intensify when considering frustrating or personally exposing aspects of working with DV.

When confronted with stories of HI in DV, police exhibited anxiety about the unpredictability and danger of HI, focusing on their vulnerability to scrutiny and blame. The risk of the individual officer being blamed for poor outcomes often became the guiding motivator. Officers did not always have accurate knowledge about HI but usually acted on the principle of “better safe than sorry”. 

It is relevant to consider that policing in the UK is based on the ideology of policing by consent [65]. In order for police to be legitimate, society must accept police as having power [66]. Findings suggest that DV poses a particular challenge to police power, and therefore police identity. Participants in this study felt that victim non-cooperation left them regularly attending DV incidents unable to prosecute or prevent crime. Indeed only 11% of arrests police made for DV related crime in 2019 resulted in a charge, with 54% assigned an outcome of evidential difficulty where the victim does not support action [67]. This may threaten participants’ self-construct as police officers with power and authority, and their constructs of victims as vulnerable and desiring police intervention. Participants spoke about their confusion around victims who were described as “vulnerable” and needing help, but also as “manipulative”, “violent” and “hostile”. This appeared to increase ingroup/outgroup tensions between police and victims. Separation was indicated in participants’ language (“we/us” referring to police, “them/they” referring to victims). 

Participants’ concepts of victimhood seemed to become distorted when there was ongoing violence in relationships. Victimhood can be understood as a socially constructed status which is attributed according to formal and informal rules [68]. Victimhood is usually constructed from ideals of suffering and interactions with social systems, especially the use of justice systems and cooperation with police [69]. When DV victims remain in violent relationships, they can violate pervasive cultural codes that present victims as having a lack of agency in their own victimisation [70]. In the present study, police officers’ experiences of DV victims did not always conform to social constructs of victimisation, for example, when police perceived victims as not taking action against perpetrators. When victimhood became distorted, some officers referred to victims as “distrustful”, “hostile”, and located some responsibility for ongoing violence with victims. This appeared to be exacerbated by the departmentalisation of police systems (part of austerity politics) which means officers only respond to the immediate violence in DV [71]. Officers remain largely unaware of outcomes, unless they attended a repeat offence, which can increase frustration and negative attitudes towards victims [72]. 

The present study specifically examined how police respond to HI in DV. Despite some inaccurate knowledge about the mechanism of BI, participants reported that they always consulted medical professionals. This suggests training in HI may not be a priority for improving police responses. Instead, analysis indicates two significant barriers to police supporting DV victims with HI: reduced resources and vulnerability to external blame. Research by Waring [73] found that police officers who experience a blame culture become distracted by anxiety and fear of punishment. Participants described DV as a particular area of scrutiny, and HI as an area of organisational anxiety for police. When faced with a HI in DV, police officers can become focused on their own vulnerability to blame, and the victim can become a threat to the officer’s safety. 

### Study Limitations and Directions for Future Research

This is the first study to investigate police responses to HI in DV victims, presenting unique insight into officers’ anxiety around HI. This study had a sample of 12, with only two female participants (16.6%), disproportionate to 30.6% of the police force who identify as female [74]. Future research could focus on women police officers. Participants in this study rarely discussed DV in members of the lesbian, gay, bisexual, and transgender (LGBT+) community. The LGBT+ community is often neglected from consideration in DV research and intervention, despite the high prevalence of DV and vulnerability of this group [75]. Research exploring officers’ beliefs and experiences policing DV in non-heterosexual relationships would help explore barriers to police supporting LGBT+ victims. 

A further limitation is the influence of self-selection and self-report bias. Eight of the twelve participants volunteered for the research. Self-selection is likely to produce participants who hold a special interest, or particularly strong views around DV. The data may represent extremes within the sample. Participants sometimes indicated anxiety within the interview about scrutiny, whether from their superiors, the media, or researchers, and social desirability bias may have influenced responses. This study is therefore only part of the overall story of police response to DV. Ideally, further research would include the views of DV victims in relation to the behaviour and professionalism of the police.

## 5. Conclusions

This study explored how police attitudes towards DV victims are constructed, and how police respond to HI in DV. Analysis suggested experiences of powerlessness and frustration produce tensions between police and DV victims, resulting in confusion around victimhood and some blaming attitudes. Analysis also demonstrated that officers’ fear about their vulnerability to external blame overtook their decision making in DV and could be a barrier to supporting victims to access healthcare. This research highlighted police vulnerability related to limited resources, scrutiny, and blame.

## Figures and Tables

**Figure 1 ijerph-18-07070-f001:**
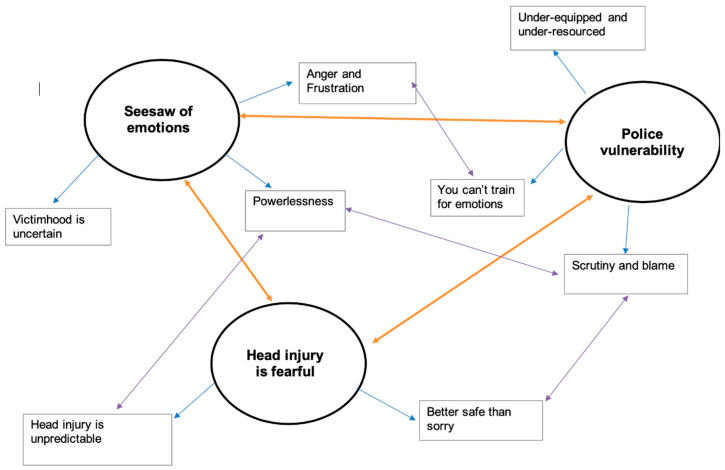
Thematic map of global themes and sub-themes.

## Data Availability

The data presented in this study are available on request from the corresponding author. The data are not publicly available due to confidentiality.

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
