# Peer review of "“If It Goes Horribly Wrong the Whole World Descends on You”: The Influence of Fear, Vulnerability, and Powerlessness on Police Officers’ Response to Victims of Head Injury in Domestic Violence"

_ijerph, 2021, doi:10.3390/ijerph18137070_

Round 1

Reviewer 1 Report

The manuscript is written very well. However, I got several concerns.

  1. Although you declared little research exploring police responses to the health needs of DV victims, I wondered if you could provide the research stream of similar study and strengthen the research gap. I believe this can enhance the contribution of your study.
  2. where is Appendix G?
  3. The results section provide a comprehensive discussion. However, it is sometimes hard to follow. I think the authors have to reorganize the flows.

Author Response

please see attachement

Reviewer 2 Report

Thank you very much for the opportunity to review this manuscript. In my opinion it is an excellent and valuable and well structured document. The topic of the study is original and very important. The methodology of the study is strong (and approved by the University Ethics Committee). The results are clearly presented and extensive discussed.  

I have only one small comment:
The reference style should be adjusted to the requirements of the journal. 

Reviewer 3 Report

1 The work is very interesting and the idea of ​​obtaining the opinion of the police force dedicated to gender violence is very timely. However, it would have been relevant to also seek the opinion of the victims in relation to the behavior and professionalism of the police. In fact, as the authors themselves acknowledge, it has been the police chief who has selected the participating professionals, perhaps with the criterion of maintaining the good image of the police, but not so much as responding to the merits of the matter at hand.

2 Subsection 4.1. it would have been more appropriate to place it in the methodology. I value positively that the researcher honestly acknowledges her difficulties and her limits, which honors her ethically. However, these reflections are perhaps more appropriate for the moment in which she made the thesis; now, in the preparation of this research article, it would have been more appropriate for me to provide answers to these obstacles and limits.

3 Subsection 4.3. on possible future research is more appropriate for a doctoral thesis than for a research article. However, it is advisable to cite the lack of research on LGRTBI people affected by gender violence and that this would have been indicated in the methodology.

4 Also, small formal issues should be taken into account, such as adding the pages to at least the most important citations. Likewise, it would be pertinent for the authors to indicate the provenance of Figure 1, especially if it is their own creation.

Round 2

Reviewer 3 Report

The authors have not followed the directions of this reviewer. For example, when I suggest that it is not pertinent to insist on the limits and difficulties of current research (it is more pertinent in a thesis) they have decided to "shorten" them. When I indicate that they specify the provenance of figure 1, they add interesting questions, but do not clearly indicate whether they are the authors. But the most important thing is that I still think that this work reflects the spirit of the previous doctoral thesis, but that it does not fit as much as a research article for a specialized journal. The results obtained from police officers previously selected by the Chief of Police and not by the authors of the investigation themselves determine the veracity of the results and that they conform to reality. Rather, they convey the desire for the good image of the police force and, therefore, do not seem sufficient to reach relevant conclusions and, of course, to suggest changes in the police force. In this sense, I understand that the authors should have incorporated into the work the opinions of the victims, which would have contrasted the "advertising" responses of the police officers. In short, in my opinion, in these circumstances, the article is not publishable without these required changes or, at least, without specifying two changes: 1º) Add in the introduction that the objective is to click on the opinion of the body, 2º) Eliminate the Suggestions from the authors, as a result of the opinions expressed by the police, on the changes that the police should develop.

Author Response

Please see attached letter 
